An assessment of fracture resistance of three composite resin core build-up materials on three prefabricated non-metallic posts, cemented in endodontically treated teeth: an in vitro study

Kumar Lalit 1 drlalitbida@gmail.com
Pal Bhupinder 2
Pujari Prashant 3
1 Department of Prosthodontics, Dr. Harvansh Singh Judge Institute of Dental Sciences and Hospital, Punjab University , Chandigarh , India
2 Consultant Maxillofacial Prosthodontist & Implantologist , Barnala, Punjab , India
3 Department of Orthodontics, Pacific Dental College , Udaipur, Rajasthan , India
Elangovan Satheesh
Electronic publication date: 2015 Feb 24
Publication date: 2015
Volume: 3
Electronic Location ID: e795
Received 2014 Dec 3; Accepted 2015 Feb 4
Copyright: © 2015 Kumar et al.
Copyright year: 2015
Copyright holder: Kumar et al.
License: This is an open access article distributed under the terms of the Creative Commons Attribution License, which permits unrestricted use, distribution, reproduction and adaptation in any medium and for any purpose provided that it is properly attributed. For attribution, the original author(s), title, publication source (PeerJ) and either DOI or URL of the article must be cited.
License URL: https://creativecommons.org/licenses/by/4.0/

Keywords: Core build-up, Non metallic post, Endodontically treated teeth, Post and core

Funding: The authors declare there was no funding for this work.

==============================
Endodontically treated teeth with excessive loss of tooth structure would require to be restored with post and core to enhance the strength and durability of the tooth and to achieve retention for the restoration. The non-metallic posts have a superior aesthetic quality. Various core build-up materials can be used to build-up cores on the posts placed in endodontically treated teeth. These materials would show variation in their bonding with the non-metallic posts thus affecting the strength and resistance to fracture of the remaining tooth structure.

Aims. The aim of the study was to assess the fracture resistance of three composite resin core build-up materials on three prefabricated non-metallic posts, cemented in extracted endodontically treated teeth.

Material and Methods. Forty-five freshly extracted maxillary central incisors of approximately of the same size and shape were selected for the study. They were divided randomly into 3 groups of 15 each, depending on the types of non-metallic posts used. Each group was further divided into 3 groups (A, B and C) of 5 samples each depending on three core build-up material used. Student’s unpaired ‘t’ test was also used to analyse and compare each group with the other groups individually, and decide whether their comparisons were statistically significant.

Results. Luxacore showed the highest fracture resistance among the three core build-up materials with all the three posts systems. Ti-core had intermediate values of fracture resistance and Lumiglass had the least values of fracture resistance.

Introduction

Aesthetics demands as well as the awareness of patients have increased over the years. A combination of new generation materials with improved clinical procedures has opened more avenues for both the dentist and the patient. Tooth-coloured materials in dentistry have progressed to the point where they can now be used confidently in almost every restorative situation.

Dental treatment and techniques have evolved from “removing the infected tooth” to “treating the infected tooth.” Endodontic therapy has transversed a meandering course, and in the present day scenario a grossly decayed tooth with a lost crown structure is effectively used to support a restoration and thereby restoring function, aesthetics, and psychological comfort for the patient. Special techniques and consideration are needed to restore such mutilated teeth to have a good prognosis (Fernandes & Dessai, 2001).

The loss of considerable amount of tooth structure makes retention of subsequent restorations more problematic and increases the likelihood of fracture during functional loading. Different clinical techniques have been proposed to solve these problems, and one such technique is the post and core. The basic objective in restoring mutilated teeth with post and core is the replacement of the missing tooth structure to gain adequate retention for the final restoration (Trabert & Cooney, 1984).

Dentistry has evolved with technological progress. With rapid research and development in the different instrumentation, post and core systems are easier than even before. Foundation restoration (as they are known today) form the base for attachments for crowns, bridges and other prosthesis (Morgano & Brackett, 1999).

In the earlier years, dowel crowns (as they were known) were fabricated to restore endodontically treated teeth where a considerable amount of tooth structure was lost. However, they were difficult to replace, as they could not be removed easily from the root canal without fracturing the root. With advances in restoration of endodontically treated teeth, the post and core system has gained popularity as an option to build the lost tooth structure. The post engaged the radicular dentin to achieve retention and the core replaced the coronal portion of the crown. This could be fabricated in metal as one piece-casted restoration or could be a separate post with a core build-up.

Various materials for posts have been introduced. To achieve the best results, the post material should have physical properties similar to dentin, be able to bond to the tooth structure and be biologically compatible (Assif et al., 1989; King & Setchell, 1990). Posts are made mostly of various corrosion resistant and rigid metals. The cast post and core has been widely used in restorations; however; its stiffness has always increased the risk of stress concentration, leading to root fracture. Custom cast post would also compromise aesthetics, as a grey tint of the metal may show through the thin root walls. The type of crown material does affect the post selection (Fernandes, Shetty & Coutinho, 2003). The growing demand for esthetic restorations has led to the development of tooth-coloured, metal-free posts which have elastic modulus comparable to dentin to prevent the tooth from fracture, potentially allowing for retreatment of the tooth and better aesthetics (Shetty, Bhat & Shetty , 2005).

Cores are built using metallic or non-metallic materials. In earlier years, amalgam was popular and in later times cements like glass ionomer and modified ionomers were used; now improved high strength composite resins are being used to build cores (Cohen & Burns, 1994). Since the advent of metal-free dentistry to achieve optimum aesthetics, tooth-coloured non-metallic post like glass fiber, quartz fiber, zirconia, ceramic have become popular. They can be used with various composite resin core build-up materials.

Composite resin core materials are used in conjunction with non-metallic posts in restoring endodontically-treated anterior teeth to achieve better aesthetics. Thus, the prefabricated non-metallic posts with composite resin core built-ups have gained popularity in the recent years. A variety of these systems are available; with this background in mind, an in-vitro study was planned to assess and compare the fracture resistance of composite resin core build-up materials with non-metallic posts in extracted endodontically treated teeth.

Materials & Methods

Forty-five freshly extracted maxillary central incisors were selected for this study. Teeth of approximately similar size and shape which were free of cracks, caries and fractures were selected. Ethical clearance was obtained from the ethical board of the institution to use extracted teeth for the purpose of this study.

Extracted teeth were scaled to remove calculus and hard debris with an ultrasonic scaler. They were then stored in saline until used. The labial and palatal surfaces were marked.

The 45 central incisors were divided randomly into 3 groups of 15 each, depending on the types of non-metallic posts used. Depending on the core build-up material, each group was further divided into 3 groups (A, B and C) of 5 samples each. Since there were 3 types of posts and 3 different core materials, there were a total of 9 subgroups having 5 samples each (Table 1).

Table 1 The samples were divided into total of 9 subgroups having 5 samples each.

			Sub groups	
Group I	Glass fiber post (Reforpost by Angelus Dental solutions Brazil).	A—Luxacore	I-A Glass fiber post+ Luxacore	
B—Lumiglass	I-B Glass fiber post + Lumiglass	
C—Ti Core	I-C Glass fiber post + Ti core	
Group II	Quartz fiber post (D.T. Light posts by RTD France)	A—Luxacore	II-A Quartz fiber post+ Luxacore	
B—Lumiglass	II-B Quartz fiber post+ Lumiglass	
C—Ti Core	II-C Quartz fiber post+ Ti core	
Group III	Zirconia post (Snow light posts by Danville)	A—Luxacore	III-A Zirconia post + Luxacore	
B—Lumiglass	III-B Zirconia post + Lumiglass	
C—Ti Core	III-C Zirconia post + Ti core	

Post systems used

(i) Glass fiber post—Reforpost by Angelus Dental solutions (Brazil)

These are glass fiber posts. They are composed of prefabricated posts made from glass fibers embedded in epoxy resins for intra-radicular reinforcement. They are nonsilanated and it was required to silanate them. A post of diameter 1.1 was selected.

(ii) Quartz fiber posts—D.T. Light posts by RTD (France)

They have unidirectional pre-tensed quartz fibers in epoxy matrix using a modified resin that wets the fibers, creating a translucent effect. It has double taper. They are nonsilanated. A post of diameter 1.2 mm was selected.

(iii) Zirconia post—Snow light posts by Danville

Zirconia posts have a high percent of Silica Zirconia fibers embedded in the polyester matrix for strength with flexibility close to natural dentin. They are high light-transmissive and white in colour, pre-silanated, and have a higher filler ratio of 60%. A post of diameter 1.2 mm was selected.

Materials used for core build-ups

(i) Composite resin dual cured core build-up—Luxacore by DMG (Dental Avenue India)

It is composed of Barium glass 69%, PyrogSilica 3% in BIS GMA matrix. Filler by weight is 72% and filler particle size is 0.02 to 4 mm. It is radio-opaque.

(ii) Composite resin Light cured core build-up—Lumiglass by RTD France (by Prime Dental India).

It consists of hybrid BISGMA composite resin. Filler by weight is 80% and filler particle size is 2–5 mm. It is radio-opaque.

(iii) Composite resin self cured core build-up—Ti-Core natural by Essential Dental Systems U.S.A.

It is composed of BIS-GMA, titanium reinforced. Filler by weight is 75%. It is radio-opaque.

Preparation and endodontic treatment of selected teeth

All the forty-five samples were sectioned 2-mm coronal to the cemento-enamel junction with a wheel-shaped diamond point on an air rotor with water spray. The teeth were prepared using a torpedo-shaped diamond point above the cemento-enamel junction, in such a way to achieve a 2 mm ferrule (Yue & Xing, 2003; Akkayan, 2004; Pereira et al., 2006) and a 1.5 mm deep chamfer finish margin (Akkayan & Gulmez, 2002).

Access opening of all 45 teeth was done with a round diamond point No. 4 (Mani, Inc., Tochigi, Japan) at a high speed with water spray. At #15 K-file was introduced into the canal to achieve patency of the canal. Pulp was extirpated with a barbed broach and constant irrigation with 5% sodium hypochlorite.

Canal length was established using a #15 K file. The working length was kept 1 mm short of the apical end. Biomechanical preparation of the teeth was done with K-files from #15 to #60 using the conventional technique. Frequent recapitulation was done to maintain patency of the canal and prevent it from getting clogged. Finally, after proper biomechanical preparation, the canal was irrigated with distilled water and stored back in saline till obturation was done.

For obturation, each of the teeth was removed from saline, and the canal was dried with paper points. The canals of all the teeth were obturated using the same standardized process. Obturation was done with gutta-percha with a non-eugenol based root canal sealer. The gutta-percha at the canal orifice was sealed with a hot burnisher; samples were stored in saline (Akkayan & Gulmez, 2002). Eugenol is shown to inhibit polymerization of composite resin (Dilts et al., 1986). Hence, a eugenol-free root canal sealer was used in the study.

Preparation of post space

The samples were removed from saline. A silicone stopper was attached to the universal drill, which was used to remove the gutta-percha and prepare the post space to a depth of 10 mm apical to the coronal dentin. The subsequent drills supplied by the manufacturer were used to further prepare the post space in order to obtain the desired length and diameter for the specific posts. The canal was irrigated with saline to remove debris.

The glass fiber posts selected were checked for their fit and length in the prepared canal. The posts were cut 13 mm from its apical end to get the required dimensions, 10 mm in the tooth (8 mm below the cemento-enamel junction and 2 mm ferrule) and 3 mm above the prepared coronal dentin, (Sirimani, Riis & Morgano, 1999) (Fig. 3).

An intra-oral periapical radiograph was taken to check the position of the post in the canal.

Etching, bonding, silanation and cementation

As instructed by the manufacturer silane was applied to the glass fiber post with a brush and air dried for 1 min. Silanation of the quartz fiber post was not required. Zirconia posts were pre-silanated, but had to be cleaned with alcohol to remove any surface impurities. The post space and the exposed part of the coronal dentin was etched and primed for 10 s with Clearfil SE, then dried. And Clearfil SE bonding agent was applied; after that, it was exposed to a light blast of air to obtain a thin layer of bonding agent, which was then light cured for 20 s. All the 45 posts were bonded with Clearfil SE (Cohen et al., 1999).

RelyX ARC resin cement was used to cement the posts in the canals. Equal amounts of base and catalyst of RelyX ARC resin cement was mixed. The canal as well as the post was coated with it. The posts were placed in the canal and held under digital pressure, and light cured for 20 s.

All the posts in various groups were cemented in the similar manner.

Composite core build-up

A preformed core former was selected for each of the samples of the teeth for the core build-up with the respective core build-up materials. The core formers were modified at the gingival end to achieve the standard dimension of the core. Luxacore (DMG Dental Avenue India, Mumbai, India) is a dual cured core build-up material.

Equal amount of base and catalyst was premixed and dispensed from the syringe into the core former. The core former with the core build-up material was placed on the post and prepared tooth surface. It was light cured for 40 s. The core formers were held in position for 5 min for complete polymerization to occur because it was a dual cured composite resin. In the similar manner, all the core build-ups were carried out for the 15 samples using Luxacore.

Lumiglass (RTD France by Prime Dental India, Maharashtra, India) is a light cured composite resin core build-up material. Ti-core (Essential Dental Systems, South Hackensack, New Jersey, USA) is a self-cured composite resin. It does not need to be light cured. In the similar manner all the core build-ups were carried out as mentioned above for the remaining 30 samples using Ti-core and Lumiglass. All above procedures were done by single operator/person.

Mounting the samples

A split mould (Fig. 1) was used to mount the teeth in autopolymerising acrylic resin. Petroleum jelly was applied on the inner surface of the split mould for easy separation of the acrylic block from the mould.

Figure 1 Photograph showing split mould for mounting samples.

Photograph by Lalit Kumar.

The teeth were mounted perpendicular to the base of the mould and embedded in the autopolymerising acrylic resin. The crown root ratio was not taken into consideration; instead, care was taken so that the cervical finish line was just above the auto-polymerising acrylic resin. All the teeth were mounted in a similar manner (Fig. 3).

Testing of the samples for fracture resistance

The acrylic block with the samples was placed on the Zwick machine for testing of the fracture resistance.

For positioning the samples on the Zwick machine a customized mounting fixture was fabricated into which the acrylic blocks fitted perfectly. The fixture also helps to position the samples in such a way that the load could be directed at 130° to the long axis of the tooth (Akkayan & Gulmez, 2002) (Fig. 2).

Figure 2 Photograph showing samples positioned at 130° on the Zwick universal load testing machine.

Photograph by Lalit Kumar.

Figure 3 Photograph showing dimensional representation of post and core foundation.

Each of the sample blocks were fixed to the base of the Zwick machine using the fixture and the tip of the plunger was made to contact the notch on the palatal surface of the core build-up. The samples were loaded at a crosshead speed of 0.5 mm/min (Fraga et al., 1998) until there was a visible or audible sign of failure in the post and core. The site at which the fracture took place was evaluated and the results tabulated. Observations thus obtained were statistically analysed.

RESULTS

The study was carried out to assess the fracture resistance of various composite resin core build-up materials with three prefabricated non-metallic posts cemented in extracted endodontically treated teeth. The 45 specimens were loaded in the Zwick machine at an angle of 130° to the long axis of the tooth. Load was applied till there was an audible or visible sign of fracture. The load at that instance was recorded as the peak load that the tooth can sustain before fracture. This was recorded for all the specimens and is listed in Table 2.

Table 2 Failure loads for all the specimens in various groups.

	Group	
Indices	I-A	II-A	III-A	I-B	II-B	III-B	I-C	II-C	III-C	
Sample size	5	5	5	5	5	5	5	5	5	
Mean	25.220	23.115	26.010	23.614	19.896	16.873	22.163	22.715	15.498	
Standard deviation ± (S.D.)	±1.4006	±3.0814	±3.3845	±2.8105	±3.2506	±1.9118	±2.2128	±3.6613	±3.3860	
Range	23.593–26.981	20.134–27.851	22.238–29.531	20.780–27.916	16.603–24.072	15.035–19.236	19.055–24.310	19.497–28.977	11.264–19.595	

These observations were statistically analyzed to comparatively evaluate the values obtained. The analysis of variance ANOVA test was applied using F distribution. It is suitable for testing the significance of difference between two or more specimens simultaneously. Since significant F does not tell us which means are different from which other means, hence we had to proceed to test separate differences by permutation and combinations through student ‘t’ test. The analysis of variance is based on a separation of the variance of all observation into parts, each of which measured variability attributable to some specific source such as internal variation of the specimen or one specimen from the other.

Student unpaired ‘t’ test was also used to analyze and compare each group with the other groups individually, and decide whether their comparisons were statistically significant as listed in Table 3.

Table 3 Mean difference between pairs of groups with its significance using students ‘t’ test.

	I-A	II-A	III-A	I-B	II-B	III-B	I-C	II-C	III-C	
I-A	–	2.105 NS	0.790 NS	1.606 NS	5.324 **	8.347 **	3.050 NS	2.505 NS	9.722**	
II-A	–	–	2.895 NS	0.497 NS	3.219 NS	6.242 **	0.952 NS	0.400 NS	7.617 **	
III-A	–	–	–	2.396 NS	6.114 **	9.137 **	3.847 *	3.295 NS	10.512 **	
I-B	–	–	–	–	3.718 *	6.741 **	1.001 NS	0.899 NS	8.116 **	
II-B	–	–	–	–	–	3.023 NS	2.267 NS	2.819 NS	4.398 *	
III-B	–	–	–	–	–	–	5.290 **	5.842 **	1.375 NS	
I-C	–	–	–	–	–	–	–	0.552 NS	6.665 **	
II-C	–	–	–	–	–	–	–	–	7.217 **	
III-C	–	–	–	–	–	–	–	–	–	
Notes.

N.S.—Non-Significant P > 0.05.

Table Value of ‘t’ for 36 degree of freedom (df).

t 0.05 = 2.02.

t 0.001 = 2.436.

S.E. D =2.88281/5+1/5=1.8231.

D 0.05 = 2.028 × 1.8231 = 3.7155.

D 0.001 = 2.436 × 1.8231 = 44630.

Largest difference is between III-A–III-C = 26.010–15.498 = 10.512.

Smallest difference is between II-A–II-C = 23.115–22.715 = 0.400.

17 differences are significant at 0.05 level.

14 differences are significant at 0.01 level.

* Significant P < 0.05.

** Significant P < 0.001.

Fracture patterns were either horizontal, oblique, some involving the core, some involving the post and tooth structure, some with debonding of post and core, and some with a combination of the above types. However, an attempt is made to classify these fractures into two groups, as shown in Tables 4 and 5. They are

1. Restorable or Salvageable Fractures

Fractures that have occurred above the CEJ, or oblique fractures that cross below the CEJ with sum amount of coronal dentin, and the oblique fracture ends in the cervical 1/3rd of the root.

2. Non-Restorable or Non-Salvageable Fractures

Fractures occurring below the CEJ with no coronal tooth structure remaining.

From Table 3 the following conclusions can be drawn as follows:

Group I-A does not differ with (Non-significant) Group II-A, Group III-A, Group I-B, Group I-C, Group II-C but differs significantly with Group II-B, Group III-B, and Group III-C at P < 0.01.

Group II-A does not differ with (Non-significant) Group III-A, Group I-B, Group II-B, Group I-C, Group II-C and differs significantly with Group III-B, Group III-C at P < 0.01.

Group III-A does not differ (Non-significant) with Group I-B, Group II-C but differs significantly with Group I-C at P < 0.05, and Group II-B, Group III-B, Group III-C at P < 0.01.

Group I-B does not differ (Non-significant) with Group I-C, Group II-C but differs significantly with Group II-B at P < 0.05 and Group III-B, Group III-C at P < 0.01.

Group II-B does not differ (Non-significant) with Group II-B, Group I-C, Group II-C but differs significantly with Group III-C at P < 0.05.

Group III-B does not differ (Non-significant) with Group III-C but differs significantly with Group I-C, Group II-C at P < 0.01.

Group I-C does not differ (Non-significant) with Group II-C and differs significantly with Group II-C at P < 0.01.

Group II-C differs significantly with Group III-C at P < 0.01.

Table 4 The number of specimens fractured as salvagable or non-salvageable in all the groups with respect to core material used.

Group	Salvagable fractures	Non-salvagable fractures	
	Nos.	%	Nos.	%	
I-A	4	26.67	1	6.66	
II-A	3	20.00	2	13.33	
III-A	4	26.67	1	6.66	
Total	11	73.33	4	26.66	
I-B	5	33.33	–	–	
II-B	5	33.33	–	–	
III-B	3	20	2	13.33	
Total	13	86.66	2	13.33	
I-C	5	33.33	–	–	
II-C	3	20.00	2	13.33	
III-C	4	26.67	1	6.66	
Total	12	80	3	20	
Grand total	36	80	9	20	

Table 5 Number of specimens fractured as salvagable or non-salvagable in all the groups respect to the posts used.

Group	Salvagable fractures	Non-salvagable fractures	
		Nos.	%	Nos.	%	
(I)	A	4	26.67	1	6.67	
B	5	33.33	–	–	
C	5	33.33	–	–	
Total: (15 = 100%)		14	93.33	1	6.67	
(II)	A	3	20	2	13.33	
B	5	33.33	–	–	
C	3	20	2	13.33	
Total: (15 = 100%)		11	73.33	4	26.67	
(III)	A	4	26.67	1	6.67	
B	3	20	2	13.33	
C	4	26.67	1	6.67	
Total: (15 = 100%)		11	73.33	4	26.67	
Grand total (45 = 100%)		36	80.0	9	20.0	

Discussion

The restoration of endodontically treated teeth has been a long concern of dentistry. These pulpally-involved teeth, which were formally considered for extraction, are now being retained with the advances in the field of endododontics and restorative dentistry. Due to loss of tooth structure and altered physical characteristics following endodontic therapy, all teeth require some form of restorative treatment.

The longevity and the success of the endodontically treated teeth depend on the procedure with which it is restored. It has been observed that pulpless teeth are more brittle than vital teeth. Also, anterior teeth are more prone to oblique forces resulting in horizontal and vertical fractures usually in the cervical third (Mclean & Gasser, 1985). If there is a conservative access opening, no carious breakdown or fracture of tooth structure and no evidence of internal or external root resorption, the tooth can survive the brunt of masticatory load (Gutmann, 1992). When there is excessive loss of tooth structure, retention for the artificial crown is required. This can be achieved by using a post and core (Morgano & Brackett, 1999). However, it should not adversely affect the load bearing capacity of the tooth. It has been indicated that the structural integrity of the tooth depends on the quality and quantity of dentin and its anatomic form (Gutmann, 1992). Both of these factors are affected when the tooth is endodontically treated, hence they may not perform their function to their fullest extent as a vital tooth. Thus, an extra-coronal restoration would be required to restore the weakened tooth. The remaining tooth structure might not be adequate enough to retain a crown, and thus a post and core is indicated. A large number of post and core systems are available with their advantages and disadvantages. Conflicting results regarding the reinforcement of the tooth due to placement of post exists making it more difficult to choose a particular system (Assif & Gorfil, 1994).

There are various core materials used in the past, such as amalgam, glass ionomer cement, modified glass ionomer and composite resin. Prepared composite resins cores have better strength than prepared glass ionomer cement cores (Stober & Rammelsberg, 2005) and prepared amalgam cores.

A variety of self-cured, light cured and dual cured composite resin core build-up materials are used in conjunction with non-metallic posts for an aesthetic restoration (Standlee, Caputo & Hanson, 1978; Dilmener, Sipahi & Dalkiz, 2006).

In this study, 45 extracted human maxillary central incisors were selected. The selection of intact natural central incisors seems to represent the best possible option to simulate clinical situation for endodontically treated anterior teeth. Previous studies have reported their use for research of various post systems (Akkayan & Gulmez, 2002; Fraga et al., 1998; Sirimani, Riis & Morgano, 1999; Raygot, Chai & Jameson, 2001). An attempt was made to choose teeth of similar root length and diameter with the help of the digital vernier calliper. The mean size of roots was 15.41 + 1.18 mm in length and 6.29 + 0.45 mm in mesio-distal width at cemento-enamel junction.

All the samples were sectioned with an air rotor 2 mm coronal to cemento-enamel junction, and a finish line of 1.5 mm deep chamfer was prepared all around the samples. A ferrule of 2 mm was prepared for all the samples (Yue & Xing, 2003; Pereira et al., 2006; Akkayan, 2004; Tan et al., 2005). This was done to simulate the natural conditions, as teeth which have fractured in the cervical one-third with insufficient coronal tooth structure remaining have to be restored with post and core so as to give retention to the artificial crown. A finish line of 1.5 mm was given to simulate the preparation for the future extra-coronal restoration (Sirimani, Riis & Morgano, 1999).

The recommended diameter of posts used for restoring maxillary central incisors is between 0.9 to 1.4 mm. Glass fiber has a diameter of 1.1 mm, quartz fiber 1.2 mm and zirconia 1.2 mm; all have been used within the above mentioned ranges.

The length of the post below the cemento-enamel junction for maxillary central incisor is 8.3 mm according to Shillingburg, Kessler & Wilson (1982). But for the ease of measurement in this study the posts were embedded to a depth of 8 mm below the cemento-enamel junction (Fig. 3). The post head was exposed 3 mm above the ferrule for retention of the core build-up (Sirimani, Riis & Morgano, 1999).

Composite resin core build-up materials have been widely used, owing to their high compressive strength, good adhesive properties, low modulus of elasticity, and their economic affordability (Piwowarczyk et al., 2002; Cohen et al., 1996). From a variety of composite resin core materials available today, three materials were selected which were widely used. Luxacore, Lumiglass and Ti-core were the three composite resin core materials chosen, each of which have different modes of curing.

The core build-ups were modified with an air rotor to give the shape of a prepared tooth so as to simulate clinical conditions. The height of the core from the cemento-enamel junction was 8 mm (Brandal, Nicholls & Harrington, 1987). It was observed that the incisal edge of lower teeth contacted the palatal surface of the maxillary central incisor 1-mm below the incisal edge of the core (Dilmener, Sipahi & Dalkiz, 2006). Thus, this point was standardized for load application by preparing a notch on the palatal surface of the core 1-mm below the incisal edge. These samples were mounted on acrylic blocks.

The load was applied on the palatal aspect at an angle of 130° to the long axis of the tooth. This was because the lower anterior teeth contacted the palatal surface of the upper anteriors at an angle of 130° to the long axis of the maxillary central incisor. Guzy and Nicholl reported that, for incisors, a loading angle of 130° was chosen to simulate a contact angle in Class I occlusion between maxillary and mandibular anterior teeth (Guzy & Nichols, 1979).

Crowns were not used in this study (Dilmener, Sipahi & Dalkiz, 2006; Burke et al., 2000; Cohen et al., 1997). It was observed that if the post and core combination has a good fracture resistance, the addition of a crown would enhance the fracture resistance of the tooth and it will be able to withstand greater forces (Kovarik, Breeding & Caughman, 1992; Kern, Fraunhofer & MueninghoffA, 1984). In this manner, the probable altering of parameters, such as material structure, shape, length, and thickness by crown restorations was avoided.

Load was applied by a Zwick universal load testing machine at a crosshead speed of 0.5 mm/min (Fraga et al., 1998). Failure threshold was defined as a point at which the sample could no longer withstand load and fracture of material, tooth or root occurred (Fig. 4). Loading to fracture represented a “worst case” scenario. Although it does not replicate what takes place in the oral environment, teeth are subjected to forces of mastication over a long period of time may cause fatigue, resulting in tooth fracture (Baldissara et al., 2006). This method of testing has been widely used by previous researchers (Guzy & Nichols, 1979; Martínez-Insua et al., 1998; Pilo et al., 2002).

Figure 4 Photograph showing fractured samples.

Photograph by Lalit Kumar.

Data thus obtained showed that Luxacore gave the highest mean fracture loads with all the three posts used.

The highest failure load was observed in a combination of zirconia post with Luxacore and lowest was observed in zirconia posts with lumiglass core build-up material. This is because zirconia is a much stronger post material than glass fiber and quartz fiber posts thus giving higher failure loads.

It was also observed that Luxacore provided only 73.33% salvageable fractures, whereas Lumiglass which is the weakest provided highest of 86.67% of salvageable fractures, and Ti-core provided 80% of salvageable fractures. Thus, the weaker the composite resin core build-up material, the earlier it will fracture at a lower load which would protect the tooth from fracturing (Kern, Fraunhofer & MueninghoffA, 1984) and thus a restoration can be done again.

Glass fiber posts showed highest percentage of salvageable fractures of 93.33%, while quartz fiber and zirconia posts both showed lower percentage of salvageable fractures values of 73.33% each.

Teeth which fractured above the cemento-enamel junction or just below the cemento-enamel junction in the coronal 1/3rd of the root with some amount of coronal dentin remaining were considered salvageable fractures (Akkayan, 2004; Sidoli, King & Setchell, 1997; Heydecke et al., 2002; Toksavul et al., 2005). There were non-salvageable fractures in the zirconia posts due to their high modulus of elasticity; because of this, greater stresses were transmitted to the tooth and thus causing it to fracture (Akkayan & Gulmez, 2002).

Thus, Lumiglass has lowest fracture resistance than Ti-core and Luxacore, but produced maximum salvageable fractures, as the core would fracture before the tooth could fracture, and failure would occur in the core rather than the tooth.

Glass fiber posts produced the maximum number of salvageable fractures. This might be related to the fact that its modulus of elasticity is very close to dentin preventing transmission of undue stresses to the tooth.

Luxacore with zirconia and glass fiber posts have a failure load greater than the biting force. However, these teeth would receive restoration, which would further enhance the fracture resistance (Akkayan & Gulmez, 2002).

The results of the above study are in consistence with results obtained by Akkayan & Gulmez (2002). They concluded that there were more salvageable fractures in glass fiber posts than zirconia posts.

The study by Fraga et al. (1998) concluded that there were more non-salvageable fractures in cast post and core rather than metal posts with composite cores. They also observed that composite resin core build-ups are preferred because they will fracture at a lower load than what is required to fracture the tooth.

In earlier studies by Fokkinga et al. (2004) showed that fiber reinforced posts had more failures than metal posts but there were more salvageable failures, whereas metal posts showed non-salvageable failures.

Composite resin core build-up materials are less stiff and more resilient than metallic cores, thus transmitting lesser stresses to the tooth. Yaman & Thorsteinsson (1992) reported that stiffer core materials increases cervical stresses and reduces apical stresses.

It was observed from the present study and the work done by other researchers, (Akkayan & Gulmez, 2002; Raygot, Chai & Jameson, 2001; Heydecke et al., 2002) that a lot of importance and emphasis is given to the strength of the posts, core and the restoration placed over them. But in the literature, the load at which fracture of the teeth (post or core) takes place is at a much higher load than that actually occurring during mastication. It may be subjected to higher load during a blow or trauma, which would lead to the fracture of the natural tooth. Therefore, the selection of the post and core should be done on the basis of tooth structure loss, type of restoration placed after the build-up and the occlusion it will be subjected to.

Conclusion

The study conducted evaluated the fracture resistance of three composite resin core build-up materials when used with three prefabricated posts cemented in extracted endodontically treated teeth. Within the limitation of the in-vitro study, the following conclusions were drawn,

1. Luxacore (dual cured composite resin) had the best fracture resistance with zirconia posts then with glass fiber posts and least with quartz fiber posts.

2. Lumiglass (light cured composite resin) had the best fracture resistance with glass fiber posts then with quartz fiber posts and least with zirconia posts.

3. Ti-core (self-cured composite resin) had the best fracture resistance with quartz fiber posts then with glass fiber posts and least with zirconia posts.

4. Luxacore showed the highest fracture resistance among the three core build-up materials with all the three post systems followed by Ti-core and the least values were observed with lumiglass.

Fracture resistance of Luxacore was best with zirconia post, lumiglass was best with Glass fiber posts and Ti-core was best with quartz fiber posts. The highest failure load was observed in a combination of zirconia post with Luxacore and lowest was observed in zirconia posts with lumiglass core build-up material.

5. (a) It was observed that maximum number of salvageable fractures occurred with Lumiglass followed by with Ti-core, and least occurred with Luxacore.

(b) It was observed that maximum number of salvageable fractures occurred with glass fiber post, while with both quartz fiber and zirconia posts same number of salvageable fractures occurred.

Additional Information and Declarations

Competing Interests

Author Contributions

Human Ethics

The authors declare there are no competing interests.

Lalit Kumar conceived and designed the experiments, performed the experiments, analyzed the data, contributed reagents/materials/analysis tools, wrote the paper, prepared figures and/or tables, reviewed drafts of the paper.

Bhupinder Pal performed the experiments, contributed reagents/materials/analysis tools, prepared figures and/or tables, reviewed drafts of the paper.

Prashant Pujari analyzed the data, contributed reagents/materials/analysis tools, prepared figures and/or tables, reviewed drafts of the paper.

The following information was supplied relating to ethical approvals (i.e., approving body and any reference numbers):

Institutional ethical committee has approved the use of extracted maxillary central incisors for the purpose of this in vitro study to be done.

Reference no. ABSMIDS/477/2005 dated 31st Aug 2005.

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
