# Peer review of "An assessment of fracture resistance of three composite resin core build-up materials on three prefabricated non-metallic posts, cemented in endodontically treated teeth: an in vitro study"

_PeerJ, doi:10.7717/peerj.795_

## Round 0.1 · original submission · Major Revisions

This in vitro evaluation to assess the fracture resistance of composite resin core build-up with non-metallic posts in extracted teeth has several major and minor issues. Apart from the several minor issues, one major issue that completely undermines the initial goals of this study is the fact that the length of the root is not accounted for during the analysis, which clearly undermines the goals of this investigation. Since it’s known that posts tend to fracture at about the apical point of it and therefore, the length of the root will have a bearing in the measured outcome. Moreover there is no need for figures 1, 3 and table 2. Therefore, this manuscript is not acceptable for publication in the current form and needs major modifications.

·

Basic reporting

As I understand it, this article reports a comparison of several core types and several direct posts in relation to occlusal forces implemented upon the system. It is a well thought out study, with useful information for the practicing clinician. However, at this stage there are several major issues that need to be corrected before consideration for publication.

Experimental design

Line 107. Preparation of Post Space.
Please clarify- it seems as though you have standardized the length of post, however, (as I understand your methods) you have not standardized the distance from the apical tip of post to apex of tooth. Is this correct? If not it should be clarified. If this is correct, I'm concerned about the fundamental issue that posts (when they fail) tend to fracture at about the apical point of the post. So, by not standardizing this length aren't you introducing another variable? In which case, it may also be possible to study this variable.

Line 93 onward.
It's not clear from your description of who did the treatment, who prepared posts, who placed posts and core? How were they calibrated? Was calibration tested? Perhaps they were all done by one person? Please clarify this point.

Line 146. Mounting the samples.
At what "acrylic height" (bone height) were these teeth mounted - was that measured and standardized? What was crown to root ratio in your artificial set-up? Isn't depth of post below bone level an important factor in withstanding pressure from occlusion? Can you make a comment about the rigidity of the acrylic vs the alveolar bone? Does this environment reasonably re-create the mouth? Does rigidity affect likelihood of fracture? A comparable situation that comes to mind is implant supported crowns vs crowns on teeth. The former are more prone to fracture because there is no PDL and less "give." Does your study create a similar difference? Please address the difference in rigidity of bone vs acrylic.

Validity of the findings

Why is there no crown placed after post-core? Is this a standard of practice and can the authors provide literature that supports this. In my practicing environment it is not a standard of care - a final crown is the standard of care. Would presence of a crown equalize all the different post and core materials? And if crown is standard care then the information from the study is less relevant.

Is there any literature to support your study model of post-core with no crown? This is a surprising approach and most colleagues in US will be confused by the decision not to include a crown. Please describe this - perhaps it is standard practice where authors practice which needs to be clearly described (and supporting literature would be important).

Additional comments

This is an interesting study and the authors should not be discouraged and should consider resubmitting. Minor issues have been noted above but a significant issue is whether post-core (without crown) is a relevant practice and is common enough to do a study on. However, the biggest concern is that the article is poorly written and needs a major re-write.

1. Line 30. "Esthetic dentistry has progressed to a point that it can completely predict the need of the patient requirements." What does this line mean? Are you talking about dentistry is so advanced now that we have ways of fulfilling all the needs of the patient? Please clarify this sentence.

2. Line 34. "virtually dead." What does this mean? Do you mean necrotic or a loss of tooth structure?

3. Line 76. "Forty five" not 45 when starting this sentence.

4. Line 76-85. Putting this section here is confusing. When reading this it seems as though the teeth already had root canal and core. The next paragraph clarifies that you actually did this and the teeth were extracted without post-core. This section should be later so that we understand how the teeth got their posts and cores.

5. Lin 166. Should be "Discussion"
More importantly, why did you combine results and discussion sections? It could be easier to follow with separate results and discussion

6. Lin 187-191. This classification is better in the methods section. It is neither results nor discussion of results.

7. Line 209-228. This section is better in the introduction. This sites previous studies and helps form a foundation for understanding the paper. It is nether results nor discussion. The discussion should compare your results in the frame of previous knowledge on the topic and include reflection and thought about your own findings.

8. Line 237-244. This section is better in the methods. This is neither results nor discussion.

9. Line 263-280. Again, this section is better in the methods.

10. Line 281-290. This is just restating and rehashing the methods section.

Thank you for the opportunity to review your study.

Reviewer 2 ·

Basic reporting

1) Forty-five freshly extracted human teeth were used for this study. Authors have mentioned that an ethical committee approval was obtained, however the manuscript did not describe why these teeth were extracted, for example periodontal reasons, decay, etc. It is important that this information to be included.
2) Statement of financial interest of the authors with the products used in the study must be included.

Experimental design

1) Manuscript did not present any power calculation as to how the authors have decided to use the sample size that they have chosen for this study?
2) Fracture resistance was tested with Zwick machine with a crosshead speed of 0.5mm/min. Information about the amount of load applied on these samples and its biological relevance is not included in the manuscript.

Validity of the findings

No Comments

Additional comments

The authors of this study have tested the fracture resistance of three composite resins (Luxacore, Lumiglass, and Ti Core) on three pre-fabricated posts (Glass Fiber, Quartz Fiber, and Zirconia posts). Study was conducted on extracted human teeth (endodontically treated) and fracture resistance was tested with Zwick machine at a 130 angle and crosshead speed of 0.5mm/min. Results indicates that the fracture resistance being highest for Luxacore, intermediate for Ti Core, and the least for Lumiglass. Also this study provides interesting insights for the clinicians in choosing appropriate combination of post and core.

---

## Round 0.2 · Minor Revisions

The authors addressed majority of the concerns but the current version still has several grammatical errors throughout. It is highly recommended that the authors completely proofread the manuscript or work with professional proof readers to rectify these errors.

Example 1: "Teeth of approximately similar size those were free of cracks, caries and fractures were selected" - it should be "Teeth of approximately similar size those that were free of cracks, caries and fractures were selected..."

Example 2: "a grossly decayed tooth with lost crown structure is used very effectively to support a restoration and return to function, aesthetics, and psychological comfort for the patient". This should be "a grossly decayed tooth with lost crown structure is used very effectively to support a restoration and thereby restoring function, aesthetics, and psychological comfort for the patient".

Reviewer 2 ·

Basic reporting

No comments

Experimental design

No comments

Validity of the findings

No comments

Additional comments

No comments

---

## Round 0.3 · accepted · Accept

The authors have adequately addressed the noted deficiencies and concerns expressed by the reviewers.